# GPs’ Perspective on a Multimodal Intervention to Enhance Guideline-Adherence in Uncomplicated Urinary Tract Infections: A Qualitative Process Evaluation of the Multicentric RedAres Cluster-Randomised Controlled Trial

**DOI:** 10.3390/antibiotics12121657

**Published:** 2023-11-24

**Authors:** Angela Schuster, Paula Tigges, Julianna Grune, Judith Kraft, Alexandra Greser, Ildikó Gágyor, Mandy Boehme, Tim Eckmanns, Anja Klingeberg, Andy Maun, Anja Menzel, Guido Schmiemann, Christoph Heintze, Jutta Bleidorn

**Affiliations:** 1Institute of General Practice, Charite University Hospital Berlin, 10117 Berlin, Germany; 2Department of General Practice, University Hospital Wuerzburg, 97080 Wuerzburg, Germany; 3Institute of General Practice, University Hospital Jena, 07743 Jena, Germanyjutta.bleidorn@med.uni-jena.de (J.B.); 4Robert Koch Institute, 13353 Berlin, Germany; 5Institute of General Practice/Primary Care, Faculty of Medicine and Medical Center, University of Freiburg, 79110 Freiburg im Breisgau, Germany; 6Department of Health Service Research, Institute for Public Health and Nursing Research, University of Bremen, 28359 Bremen, Germany

**Keywords:** antibiotic resistance, urinary tract infection, primary care, process evaluation, acceptability, antibiotic stewardship, qualitative research

## Abstract

Urinary tract infections (UTIs) are among the most common reasons patients seeking health care and antibiotics to be prescribed in primary care. However, general practitioners’ (GPs) guideline adherence is low. The RedAres randomised controlled trial aims to increase guideline adherence by implementing a multimodal intervention consisting of four elements: information on current UTI guidelines (1) and regional resistance data (2); feedback regarding prescribing behaviour (3); and benchmarking compared to peers (4). The RedAres process evaluation assesses GPs’ perception of the multimodal intervention and the potential for implementation into routine care. We carried out 19 semi-structured interviews with GPs (intervention arm). All interviews were carried out online and audio recorded. For transcription and analysis, Mayring’s qualitative content analysis was used. Overall, GPs considered the interventions helpful for knowledge gain and confirmation when prescribing. Information material and resistance were used for patient communication and teaching purposes. Feedback was considered to enhance reflection by breaking routines of clinical workup. Implementation into routine practice could be enhanced by integrating feedback loops into patient management systems and conveying targeted information via trusted channels or institutions. The process evaluation of RedAres intervention was considered beneficial by GPs. It confirms the convenience of multimodal interventions to enhance guideline adherence.

## 1. Introduction

Urinary tract infections (UTIs) are among the most common bacterial infections in general practice and are a frequent reason for seeking health care [1,2]. Worldwide, in 2019, there were an estimated 404.61 million cases of UTIs [3]. 

In Germany, the one-year prevalence of UTIs was 5.8% in 2015 (men: 2.5%, women: 9.2%), with GPs serving as the primary diagnosing professional for UTIs in women [4]. Several studies have described a very high likelihood of an antibiotic being prescribed when a UTI is diagnosed [5,6,7]. In primary care, UTIs are considered one of the top ten reasons for antibiotic prescription [8], and most of the consumed antibiotics are prescribed in primary care [9]. Antibiotic use can disrupt the gut and vaginal microbiome, thereby fostering drug-resistant bacteria [10]. Further, antibiotic resistance after UTI treatment increases, especially after multiple courses [11]. The economic repercussions of antimicrobial resistance (AMR) include a severe escalation of healthcare costs due to increased hospitalisation rates and medication usage [12].

Health technology assessment studies in the United Kingdom show that adherence of physicians to clinical practice guidelines is a strong predictor of clinical outcomes [13]. Clinical guidelines for uncomplicated UTIs in patients with light or moderate symptoms recommend prescribing antibiotics or symptom-based treatment [14,15]. First-line treatment options are: fosfomycin-trometamol; nitrofurantoin; nitroxoline; pivmecillam; or trimethoprim. Fluoroquinolones and cephalosporins are not recommended [14,16]. 

However, GPs’ adherence to guidelines in Germany is low, and the use of second-line antibiotics, especially quinolones, is high [1,17]. Nonadherence to guidelines at a general-practice level is linked to a severe increase in unnecessary antibiotic prescription [18] and an increase in AMR [19]. Effectively increasing guideline adherence requires more than simply disseminating knowledge. Instead, strategies should incorporate training, supervision, technology-based reminders, prescription feedback comparing individual performance to peers, information on local antimicrobial resistance patterns of UTI pathogens, and patient education [20,21]. Multimodal interventions are generally considered more effective than single interventions [22,23,24] due to their ability to address diverse target groups [23], exploit synergistic effects [25], and incorporate both public health-informed macro determinants and patient- and practice-specific micro determinants [26].

GPs play a pivotal role in implementing guidelines to combat AMR [27]. Therefore, it is essential to address the specific barriers they face in their daily practice, such as intense workloads, specific attitudes or knowledge gaps, patient expectations and inconsistencies in guideline recommendations [20,28,29,30,31]. Furthermore, information should be easy to understand, concise and delivered by credible and economically independent actors, such as universities or national public health institutions [32].

The primary aim of the RedAres (REDuction of Antibiotic RESistance) study is to reduce the proportion of second-line antibiotics through a multimodal intervention. This intervention consists of the following elements: (1) information on current guideline recommendations for UTI; (2) information on regional resistance data; (3) individual feedback on prescription behaviour; and (4) feedback on prescription behaviour in comparison to peers (benchmarking) [33].

Based on the exploratory qualitative evaluation of GPs’ perspectives on interventions to optimise antibiotic prescribing [20], the process evaluation of the RedAres study aims to identify the determinants of decision-making for antibiotic prescribing, GPs’ perceptions of the RedAres interventions and perspectives on the potential for implementation of interventions to enhance guideline adherence.

## 2. Materials and Methods

### 2.1. Study Design

The study was integrated as a process evaluation into the RedAres project, a multicentric cluster-randomised controlled trial that aims to evaluate aggregated data of uncomplicated UTI cases from 128 GP practices (64 interventions and 64 controls) in four German regions (Berlin–Brandenburg, Bavaria, Baden–Wurttemberg and Thuringia). The reduction in second-line antibiotic use in UTIs after 12 months was the primary goal of the RedAres project [34]. The interventions consisted of four modules: First, information material, including a guide for UTI management, a pocket card with a brief summary of the guideline information, as well as multilingual flyers and posters for patients. Second, representative regional and national resistance data from UTI samples in ambulatory care developed by the Robert Koch Institute. Third, individual feedback on GPs’ prescription behaviour, and fourth, feedback as compared to the average prescriptions of the other participating practices (benchmarking) [33,34].

The process evaluation consisted of qualitative interviews and a quantitative survey. The results of the quantitative survey are reported elsewhere.

We used the COREQ (Consolidated criteria for reporting qualitative research) checklist by Tong et al. for quality assurance [35]. Data were analyzed based on the methodology of qualitative content analysis by Mayring [36,37].

### 2.2. Participant Selection

We opted for a non-probability sampling based on the voluntary response of the participating GPs who were approached during the final visit of the RedAres trial. An incentive was proposed for an interview independent of the length of the interview. All participants consented to be contacted by the interviewer via an informed consent form. They were contacted via email or, in isolated cases, by a prior call. After the first interviews, we actively sought participation from underrepresented regions and genders to reach maximal variability regarding these characteristics. 

### 2.3. Setting

The interviews were carried out in German. All interviews were conducted online via video call. Only the interviewer and the interviewee were present. We audio-recorded all the interviews.

### 2.4. Data Collection

An interview guide was prepared based on the domains of the interventions. The interview guide was developed to explore the subjective usability of the study’s interventions, their feasibility, the GP’s acceptance and thoughts towards future applicability and implementation of the interventions in everyday practice. The interview guide was piloted in four interviews with GPs and researchers at the Institute for General Practice at Charité. We adapted the interview guide in an iterative manner based on the evaluation of the first four interviews. We did not carry out repeated interviews.

Field notes were taken during the interview and served as a memory aid for the interviewer during the conversation. 

The interviews were conducted between 17 August 2022 and 30 September 2022. We carried out interviews until data saturation after 19 interviews was achieved. 

Five of the interviews were transcribed by the interviewer. The remaining 14 interviews were transcribed by a scientific transcription office. 

### 2.5. Research Team and Reflexivity

The study was supervised by AS, CH and JB. The interviews were conducted by PT, a female third-year medical student. All researchers had previous experience with qualitative research, either through previous research projects or through the exchange with the qualitative research network at the Institute of General Practice. AS, CH and JB are GPs and Public Health researchers and were involved in the implementation of the RedAres Trial in Berlin–Brandenburg or Thuringia. JB has led a qualitative exploratory study on GPs’ perspectives on antibiotic use in UTIs before the RedAres trial.

There was no personal or other relationship between interviewer and respondent other than email contact to make the appointment. The interviewer presented herself as a medical student and student assistant within the RedAres project.

### 2.6. Data Analysis

The interviews were coded by PT and counter-coded by JG, AS, and ZF (another medical student). Ten out of nineteen interviews were coded twice. 

Using a mixed inductive–deductive approach, a code system for qualitative content analysis was created (see Appendix A).The inductive categories in the coding tree consisted of five major blocks. The evaluation of the technical and organisational feasibility of the RedAres study, perception of the RedAres intervention components regarding acceptability, usability and transferability to regular care and the evaluation of determinants of decision-making. Deductive categories can be summarised as follows: information gain; confirmation of prescribing; reassurance on prescribing; reflection; and influence on prescription patterns. An overview of the inductive and deductive categories is displayed in the codebook (Attachment 1). MAXQDA 2022 was used for data management and analysis. The following participant quotations are presented to illustrate the findings. They can be identified via participant numbers.

### 2.7. Ethics

Ethics approval was obtained at the Ethics Committee of the Medical Faculty, University of Wuerzburg, in November 2019, under the number 20191106 01. Data safety complies with the European and Charité regulations. All respondents signed an informed consent form prior to the interview. In January 2020, the RedAres study was registered at the Trial registration site DRKS under the trial registration number DRKS00020389.

## 3. Results

Thirty-two of the 64 RedAres intervention practices gave their consent to be interviewed for a qualitative process evaluation interview within the RedAres study. Thirteen of them did not participate; three due to time constraints, and ten practices were unreachable by phone or email. We interviewed 19 GPs: 11 males and 8 females. Eight came from Bavaria, four from Baden-Württemberg, five from Berlin–Brandenburg and four from Thuringia. The mean age was 52 years (IQR 41–61). Of the 19 practices, 4 were in villages, 6 in small cities, 4 in medium-sized cities, and 5 in metropolitan cities. The interview length was between 46 and 87 min, with a mean duration of 62 min. Data analysis revealed three key domains for further exploration: general decision making factors; overall perceptions of the RedAres study; and perspectives on RedAres intervention implementation. The overview of our main results is displayed in Figure 1.

### 3.1. Determinants of Decision Making

GPs identified various factors influencing their decision-making process. Central results with illustrative citations are displayed in Table 1. Many GPs emphasised the importance of detailed clinical information, medical history, diagnostic tests, drug interactions, and therapy recommendations. Professional experience and routine played a crucial role in guiding clinical decisions. GPs with less experience relied on information retrieval to address uncertainties. 

The collegial exchange about AMR influenced decision-making for some GPs. Some engaged in occasional discussions with colleagues during training or informal exchanges, covering both specific cases and general AMR topics. Others exchanged knowledge with laboratory physicians or specialists. Time constraints and a perceived lack of benefits hindered collegial exchange. Regular interactions were common among physicians working in joint practices or with training assistants. Obstacles to effective decision-making included limited access to updated information and concerns about the influence of pharmaceutical companies on available data. The overwhelming amount of information and difficulty filtering out essential details were also mentioned. Lack of time for thorough diagnostics and guideline-based therapy hindered informed decision-making, as did older age. Some physicians expressed concerns about losing patient loyalty if they restricted antibiotic prescriptions.

### 3.2. Perspectives on the RedAres Interventions

In general, the RedAres study was perceived as positive, especially regarding its aim and the integrability of the interventions into the daily practice. 

In the following, we describe GPs’ perspectives on the four interventions. Central results and quotes are resumed in Table 2. 

#### 3.2.1. Information Material on Guideline Recommendations

GPs perceived the information material as useful; they used it for information refreshers, reassurance, and to gain knowledge on UTI management. However, some GPs considered UTIs easy to treat and did not see the need for additional material. Those GPs saw no need to change their prescription habits. 

The information material was used for self-education, educating trainees and students, communicating with patients, and legitimising non-antibiotic treatments. Multilingual material was particularly beneficial. Some GPs found it useful to have material on hand for patients.

#### 3.2.2. Regional Resistance Data

GPs generally considered information on national and regional resistance to be very important. They expressed interest in the resistance situation and regional differences and reported gaining knowledge about these patterns. The intervention served as a reminder of different antibiotics that can be used without increasing resistance development. Several GPs adjusted their prescribing habits, switching from second-line or resistance-prone antibiotics to less-resistant first-line options. Some GPs felt validated in their prescribing practices when resistance data and recommendations aligned with their usual habits.

Not all GPs found resistance data relevant to their daily practice. Some found it interesting but did not directly change their prescribing habits. Some GPs reported reflecting on their prescribing but stuck to their habits based on past positive experiences with specific antibiotics. Others considered that ambulatory care did not contribute to antibiotic resistance and found resistance data more relevant for hospital physicians.

#### 3.2.3. Prescription Feedback

Prescription feedback was well-received by GPs and confirmed their prescribing practices. Even though it did not lead to immediate changes, the feedback provided reassurance and prompted some GPs to reassess their prescribing habits. Several GPs reported that, especially with common consultations, prescribing a certain drug becomes routine and that feedback was helpful to break those routines and to (re)align prescriptions to current guideline recommendations. GPs were familiar with prescription feedback, but their previous experience was limited to economic evaluations. They expressed dissatisfaction with the documentation of stand-by prescriptions, which might have inflated antibiotic prescription numbers. 

#### 3.2.4. Benchmarking

GPs generally viewed benchmarking favourably, finding it informative and motivating. Some received confirmation of their prescribing practices, whilst others were reminded of first-line antibiotics they had overlooked. Sometimes, those antibiotics were not included in the antibiograms sent by a laboratory to identify the susceptibility or resistance of bacterial pathogens. Some GPs did not see the value in benchmarking against other practices, preferring comparisons to optimal therapy recommendations. Others criticised the lack of comparability between practice profiles and found the comparisons unhelpful or even offensive due to limited opportunities for response. 

#### 3.2.5. Perception of the Intervention Format

GPs appreciated the clear presentation of feedback and information material. They requested short, easy-to-read formats that allowed them to quickly access specific information during consultations. Summarised and easy-to-understand formats, such as decision trees, were particularly popular. 

### 3.3. Promoting Factors and Barriers for Implementation

Incorporating regional resistance data and prescription feedback into routine care was generally welcomed by GPs. Perspectives on promoting factors and barriers to implementation are summarised in Table 3. 

Many GPs requested regular updates on resistance data, suggesting distribution through public campaigns, universities, or the Association of Statutory Health Insurance Physicians. Others preferred receiving information from collaborating laboratories due to their perceived credibility. Several GPs expressed interest in receiving updates on current regional resistance patterns for other infectious diseases, particularly for common clinical presentations. They suggested sending updates every six months or once per year and breaking down regions into smaller areas to make the information more relevant. GPs welcomed the incorporation of prescription feedback into routine care, especially when accompanied by treatment recommendations. They offered numerous suggestions for optimising feedback implementation. One proposal was to develop easier solutions, such as automatic prescription data recording and feedback integration as a plugin into practice management software.

GPs found collecting routine data to be a straightforward and efficient method for gathering valuable information whilst adhering to data privacy regulations. The interviewees expressed general openness to using routine data for research to improve guideline adherence. Provided their anonymisation, many GPs were comfortable with their prescription data being used for research or feedback purposes. Whilst supportive of integrating regional resistance data and prescription feedback into routine care, some GPs raised concerns about comparing practices due to patient population disparities. They also recognised the potential of secondary data analysis in practice data management systems but highlighted that utilisation was hindered by data privacy, administrative and technical barriers. Some GPs raised concerns about the use of routine data for feedback as missing information that could hinder a full understanding of the treatment rationale. Skepticism was expressed regarding the feasibility of implementing guideline-compliant therapy recommendations due to potential conflicts between guideline adherence and cost-effectiveness, as adhering to guidelines may not always align with the most economical treatment options. Additionally, GPs voiced concerns about the potential misuse of routine data by pharmaceutical companies and the implementation of regulatory control mechanisms. They emphasised that feedback should not be punitive.

## 4. Discussion

In the qualitative process evaluation of the RedAres study, acceptability and usability of the multimodal intervention to enhance guideline adherence was confirmed in primary care settings. These findings are in line with other studies that have shown that multimodal interventions have a positive effect on guideline-adherent prescribing in primary care [38,39]. GPs considered all components of the study beneficial and estimated them as helpful for knowledge gain and confirmation of prescribing behaviour. Information material and resistance data were used for patient communication and teaching purposes. Feedback and benchmarking were considered helpful in breaking routines in clinical workups and to reflect on prescribing behaviour. 

Our results showed that effective information materials for antibiotic prescribing should be clear, concise, and user-friendly and produced by trustworthy institutions. References to primary research or divergent recommendations that can cause mistrust should be avoided. Our results are in line with current implementation research on guideline adherence [28]. Information materials based on reliable, up-to-date evidence can enhance self-efficacy among doctors, which is crucial for behavioural change, particularly for those who believe they lack the necessary knowledge to change their practice [40,41]. Educational interventions are a cost-effective and efficient measure to reduce antibiotic prescriptions in primary care [31]. The use of information material for patient communication or training needs to be acknowledged when implementing these formats. Hence, co-design approaches could facilitate the development of formats that meet the requirements of doctors and patients [42].

The provision of resistance data for the outpatient setting to improve guideline adherence has not been implemented before [33] and was generally considered useful. Although the share of antibiotic prescriptions used for outpatient care in Europe ranges from 70 to 85% [43,44,45], most of the interviewed GPs saw the responsibility for AMR within the hospital and doubted their own responsibility. This estimation could be attributed to the fact that secondary care providers more frequently encounter complex infections, increasing their exposure to antibiotic resistance [46].

Enhancing interprofessional collaboration between human and veterinary medicine could improve the precision of local resistance patterns, amplify the information value of resistance dynamics [47], and promote evidence-based practices [48].

Despite their self-assessment of guideline adherence, GPs’ antibiotic prescribing practices did not always align with established guidelines. This finding corroborates previous research by Davis et al., suggesting that physicians’ self-assessment of their behaviour may not be reliable [49]. Physicians’ self-perception significantly influences the effectiveness of interventions aimed at improving prescribing behaviour [49]. UTIs were often considered too straightforward to warrant reflection. Thus, self-reflection and peer group opinions could be key factors in sustaining positive changes in prescribing practices [50]. 

Whilst feedback highlighting poor performance demonstrated effectiveness in our study, aligning with behaviour change models [40,51], other trials failed to achieve sustained improvement. This may be attributed to the existence of effective stewardship programmes in routine British healthcare, as well as the limitations of individual goal setting and verbal feedback in a real-world pragmatic RCT setting [52]. 

Implementing delayed prescriptions, shown to reduce antibiotic intake, could further enhance the impact of feedback [23]. A digitalised health system linking patient data to pharmacy records is essential for accurate prescribing data. Specific surveillance accounting for medical specialties would provide better insight.

An example of successful multisectoral cooperation and surveillance system to reduce AMR is the Swedish Strama programme [53]. A wide variety of views exist on the benefits of benchmarking, ranging from motivating to potentially intimidating. Social norm feedback has been proven effective to reduce antibiotic prescriptions [54,55], and should not be underestimated. The validity of benchmarking might be improved by clustering practices with similar patient structures and its effectiveness enhanced by prioritising those having a higher rate of antibiotic prescriptions. Factors influencing high prescription rates are patient morbidity and ethnicity, practice structure, and physician characteristics [56]. The integration of automated recommendations into patient management was widely accepted and has shown its efficiency in recent research [57]. However, in Germany, the plethora of patient management systems and concerns about their reliability for this scope challenge implementation into routine practice. 

### 4.1. Potential Drivers for Implementation

It is of major importance to enhance the understanding of AMR, including the public health perspective in structured mandatory courses for students, early career, and established GPs. Massive open online courses, such as the one offered by the RAI Project on rational antibiotics in primary care, are best practice examples [58]. According to the accumulation model of change [59], the spread of information via different channels, such as newspaper articles, scientific material, or feedback loops, along with the relative credibility of these different sources, drives change.

Interprofessional collaboration with laboratories was considered trusted and valuable and could, therefore, be seen as a pathway to convey updated information on regional resistance data. The adaptation of antibiograms according to current guidelines and regional resistance patterns in ambulatory care might increase trustworthiness in the laboratory and self-efficacy, an important driver for change in behavioural theory [41].

To make routine data a meaningful basis for decision-making, sentinel practice-based surveillance with regular microbiological analysis of urinary samples of complicated and suspected UTI could improve overview of resistance patterns in ambulatory care. As an alternative, regular cross-sectional evaluation studies in antibiotic resistance could be implemented [60].

### 4.2. Potential Barriers to Implementation

GPs expressed concerns regarding the difficulty in identifying objective, evidence-based information and the perceived influence of pharmaceutical companies on prescribing practices. This aligns with research demonstrating that exposure to pharmaceutical information is associated with increased prescription rates, higher costs, and less guideline adherence [61]. As previously reported, the overwhelming amount of information material made it challenging for GPs to identify reliable sources [62]. Tools to filter relevant information for GPs should be developed. 

Apprehension was expressed regarding the potential of monitored antibiotic prescriptions to be accessed by pharmaceutical companies or misused as a tool to sanction doctors.

A reason for this fear could be that regulatory bodies in Germany call GPs to be cost-effective and sanction above-average use of expensive drugs, including antibiotics, whilst guideline adherence is out of their scope. Therefore, in clinical practice guideline-adherent prescribing needs to be balanced with economic prescribing. This inhibits a physician’s ability to make guideline-adherent antibiotic prescribing decisions [32]. Hurdles shifting the focus from patient care to cost-driven decisions are well described in the literature [29]. Alignment challenges between cost and patient-centeredness are more common in decentralised systems like Germany or Belgium, and go along with difficulties in scaling-up effective measures [63]. Numerous government policy interventions aim to reduce antibiotic prescribing and AMR [64], e.g., in Taiwan by means of prohibition [65] or in Sweden, where pay-for-performance measures have shown to have a positive impact, with a sustained and long-term increase in narrow spectrum antibiotic prescribing [66]. However, financial penalties seem to have the largest effect in decreasing antibiotic prescriptions, followed by bonus systems [67]. Strategies that combine financial penalties with rewarding measures, potentially integrated through gamification, hold promise in enhancing guideline adherence [68].

### 4.3. Including a Public Health Perspective in Clinical Decision Making

Linking individual clinical practice to AMR as a global health problem seems to be a challenge for most physicians.

Whilst there is a general awareness of the growing AMR problem, there is also skepticism about GPs’ individual responsibility for it. This aligns with our interview findings, where GPs expressed responsibility at an individual level but not on a public health scale. Similarly, a study by Simpson et al. found that GPs questioned the evidence linking GP antibiotic prescriptions to AMR [69]. Gutscher et al. found that treatment decisions in general practice are usually based on the well-being of individual patients and less on the public health concepts such as the avoidance of resistance. This prioritisation is reflected in the choice of broad-spectrum antibiotics which are perceived as more effective and better for the patient [70].

Difficulties in seeing AMR as a serious threat in primary care are explainable through the cognitive dissonance in the health belief model [71], which can explain the gap between cause and effect specific to antibiotic resistance. The sense of ownership for tackling AMR is not as prevalent among primary care physicians, as secondary care and hospital doctors are often seen as bearing a greater responsibility for its development and management [46]. An increase in public health awareness for AMR among GPs could be achieved through different means: First, by conveying resistance data at the regional level to all concerned actors. Collective responsibility for AMR as a public health problem could promote interprofessional collective regional efforts (groupthink theory) [72]. Second, by conveying individualised information material about the role of GPs in the context of AMR individuals, clear and manageable responsibilities could be communicated, thereby enhancing self-efficacy and reduce inertia (inverse social facilitation theory) [40,41,73]. Third, self-efficacy could be further enhanced by integrating GPs into the process of interprofessional guideline development [40,41]. A good example of such effort is the guideline on uncomplicated UTIs, which is currently revised by representatives of nine professional associations, including GPs [15].

### 4.4. Strengths and Limitations

The limitations of the study include a possible social desirability bias of interviewed GPs [74]. Moreover, there might have been a selection bias towards more engaged and guideline-adherent doctors in the RedAres Study and, furthermore, among those interested in participating in a qualitative interview. To reduce this bias, incentives of 105€ per interview were given to the study participants.

The fact that delayed prescriptions have not been particularly differentiated from other antibiotic prescriptions might eventually distort the accuracy of prescription feedback and partly explain criticism towards feedback. Further, the structural differences between the various practice profiles may have contributed to unbalanced feedback, which may have contributed to a negative perception of the intervention. 

Our qualitative data can be linked to our quantitative evaluation, which will be published elsewhere and will increase the external validity of our study results. However, a triangulation of our data was not feasible.

## 5. Conclusions

In the qualitative process evaluation of the RedAres study, we were able to show that GPs experienced a gain of knowledge through information material, regional resistance data and feedback, which was linked to a self-perceived change towards rational antibiotic prescribing. This confirms the convenience of multimodal interventions to enhance guideline adherence. Information on regional resistance data and feedback on prescribing behaviour matching with individual prescription behaviour seems to foster self-efficacy, increased awareness, and induce maintained guideline adherence.

## 6. Recommendations

Future research should focus on translating our findings into practical applications for routine clinical practice. This includes exploring methods for utilising and processing routine data to generate feedback and integrating this information into existing information streams, such as patient management systems and resistogram data from collaborating laboratories. Additionally, the effectiveness of gamification as a tool to enhance the acceptability of information and feedback should be evaluated. For implementation research efforts, we recommend collaborating with reputable and influential institutions, such as funding agencies, who can integrate information and feedback into routine economic impact recommendations and facilitate long-term implementation.

## Figures and Tables

**Figure 1 antibiotics-12-01657-f001:**
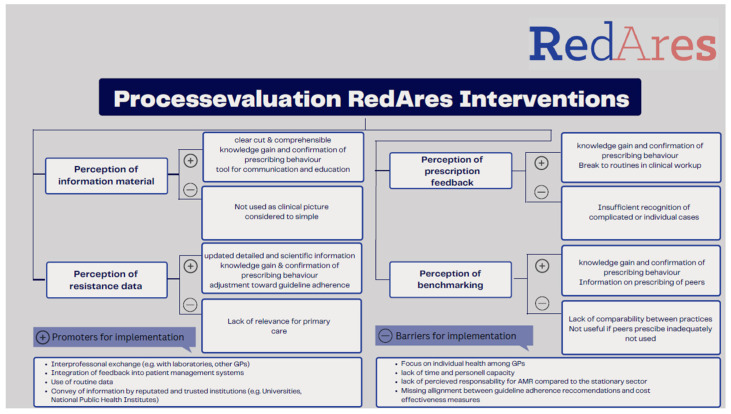
Summary of perceptions on intervention materials on implementation.

**Table 1 antibiotics-12-01657-t001:** Determinants of decision making. Central results with illustrative citations.

Subcategory	Citation
Perceived competence	*“Basically, I need a lot of routine and then I feel safe.” (10-by_int_63: 29)* *“Sometimes you feel insecure there […] confidence comes when you have evidence-based guidelines, where you can say, okay, this helps and I’m doing everything right.” (17-by_int_09: 28)*
Collegial exchange	*“We are lucky enough to have such a round table of further education in our region. […] There’s a relatively large amount of exchange in collegial discussions, and […] questions like these [about antibiotic prescriptions] also come up in the round afterwards” (4-ber_int_25: 31)*
Time constraints	*“If I treated all my patients according to the guidelines, they would be busy with medicine all day. So [...] you can read through it, that’s what we do, but I can’t really sign off on treating patients absolutely according to guidelines.” (5-bw_int_49: 21)*
Quality of information	*“With some recommendations, one has nevertheless somewhat the hidden suspicion if there are industrial interests [...] behind it. One will then perhaps sometimes be a little more defensive.” (11-bw_int_60: 37)* *“You also get a lot of information that is almost too much [...], then you can get quite lost in all this stuff [...].” (15-th_int_34: 42)*

**Table 2 antibiotics-12-01657-t002:** Perspectives on the RedAres intervention. Central results with illustrative citations.

Category	Subcategory	Citation
Information material	Knowledge and confirmation	*“Yes, it was a good refresher for me.” (15-th_int_34: 18)* *“some drugs like [...] Pivmecillinam were unknown to me.” (9-by_int_10: 35)*
	Education and patient communication	*“I also gave [the information material] to the training assistants and talked to them. And that was also good feedback for both of us.” (2-ber_int_05: 15)* *“Materials for patients are also important. Particularly when they are faced with unfamiliar therapeutic decisions [...] it is nice to have an argumentation paper with the stamp of the [university clinic] or a larger institution.” (2-ber_int_05: 25)*
Regional resistance data	Knowledge and confirmation, breaking routines	*“That was an eye-opener. [...] I was very grateful for this information, because I really wouldn’t have gone to the Robert Koch Institute website on my own.” (14-bw_int_57: 25)* *“What came from the Robert Koch Institute is also well in line with our practice”. (13-by_int_69: 51)* *“Especially with uncomplicated infections, you don’t think too much about what you prescribed, what you prescribed last time. [...]. But it’s interesting to have that presented to you and to see, oh, you took this (decision) once, which you actually didn’t want to take anymore.” (4-ber_int_25: 51)*
	Relevance for practice	*“Resistance avoidance is a relevant point [...] I just think that the regional resistance data do not influence every decision.” (16-bw_int_56: 38)* *“I think it’s more relevant in the stationary setting, because we often don’t have these decisions that require antibiotic stewardship [...] I think we must set other priorities” (16-bw_int_56: 34)*
Prescription feedback	Knowledge and confirmation	*“You think: Well, we’re doing everything right, and so on. But it’s nice to see it again in print and then finally quoted in some kind of diagram.” (2-ber_int_05: 65)* *“that’s what makes reflection possible in the first place [...] it [...] provides an opportunity for discussion and reflection.” (12-th_int_36: 65)*
	Experiences and hurdles	*“I prescribe [antibiotics] and say: [...] the urinary tract infection is so strong you have to take it now. [...] or they just get it as a backup for the next time.” (5-bw_int_49: 49–53)* *“We get feedback [from the Association of Statutory health insurance Physicians], not about the resistances and the right antibiotic, but about [economic] prescribing behaviour.” (17-by_int_09: 54)*
Benchmarking	Knowledge and confirmation	*“Actually, I think that’s quite good, because you actually compare yourself a bit. [...] So if I were completely off the mark, I would ask myself: What am I doing differently? (6-th_int_44: 73)* *“The comparison has somehow also shown that we have actually done quite well [...] encouraging me to continue in this way.” (19-by_int_65: 73)*
	Experiences and hurdles	*“And I think that’s problematic in part because you can’t compare the practices with each other”. (17-by_int_09: 54)“I don’t necessarily always have to compare myself with others.” (14-bw_int_57: 45)*
Intervention format	Experiences and hurdles	*“I need a clear recommendation from which I can derive a clear recommendation for the individual case. [...] I also don’t need a lot of justifications or [...] references to studies. Basically, I need a mini guideline that I can use.” (10-by_int_63: 29)*

**Table 3 antibiotics-12-01657-t003:** Perspectives on implementation: central results with illustrative citations.

Category	Subcategory	Citation
Promoting factors	Information streams	*“So [with] local resistance situations [...] I would hope that it would be mirrored more often in the future* via *the laboratory.” (19-by_int_65: 61)*
	Presentation of information	*“Generally, I think that this feedback is very useful. Especially if [...] coupled with [...] information that is short and [...] that considers essential aspects of recent developments or guidelines.” (18-ber_int_19: 74)* *“You can do this anonymously and simply say: [...] we have observed in your region that it is like this and like that [...]. Now, if someone points the finger at a particular colleague and says: “You did this and that wrong”, then, of course, it’s not comfortable on a personal level.” (4-ber_int_25: 83)*
	Use of routine data	*“So, more transparency and more [...] data collection. [...] We need the routine data, [...] how else are we really going to make serious scientific progress”. (18-ber_int_19: 98–100)* *“I mean, in the end you have to say that if you take the individual data into account, you can of course subdivide them a bit more precisely [...].” (3-by_int_66: 65)*
Barriers	Comparability	*Therefore, I find it relatively difficult to compare the practices with each other [...] So I wouldn’t draw any information from it if it said that I prescribed something completely different than all the other practices.” (4-ber_int_25: 65)*
	Data misuse	*So [the possibility to access the data] should really only be available to independent research institutions [...]. As soon as there are any possibilities that this could drift into pharmaceutical companies [...] then of course it is very problematic.” (12-th_int_36: 81)*
	Financial penalties	*“We always get the information from the Association of Statutory Health Insurance Physicians about which drugs we prescribe too much. But there, it is always associated with severe penalties [...]. Of course, I don’t think that’s such a good thing.” (10-by_int_63: 75)*
	Conflict of interests	*“If you always [...] list drugs that are not in discount contracts or that are not among the cheaper ones, then we get audit problems because we are required to meet certain targets for those in primary care.” (9-by_int_10: 17)*

## Data Availability

The data presented in this study are available on request from the corresponding author. The data are not publicly available due to data safety restrictions agreed with the interview partners.

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
