# Peer review of "GPs’ Perspective on a Multimodal Intervention to Enhance Guideline-Adherence in Uncomplicated Urinary Tract Infections: A Qualitative Process Evaluation of the Multicentric RedAres Cluster-Randomised Controlled Trial"

_antibiotics, 2023, doi:10.3390/antibiotics12121657_

Round 1
Reviewer 1 Report
Comments and Suggestions for Authors
The study proposed by Schuster A. et al. presents great relevance. Understanding the perspective of primary care physicians on multimodal intervention to improve adherence to guidelines in uncomplicated UTI has a direct impact on quality of care, prevention of complications, efficiency of the health care system, and quality of life of patients. In addition, it contributes significantly to the body of scientific knowledge, providing valuable information for future research and health policy.
I have reviewed the study with great interest and have identified some observations that I consider pertinent to improving its quality. These observations are detailed below:
Introduction: To provide data on urinary tract infections globally, followed by Germany, and their impact on patients' quality of life and health care costs. To explain the existence of clinical guidelines for UTI management and their importance in medical practice.
Emphasize the importance of understanding the perspective and challenges of general practitioners, who play a crucial role in the management of UTI in primary care. Justify why a multimodal intervention could be effective in improving adherence to guidelines, considering approaches that go beyond simple information dissemination.
Methodology:
Add study design (revise acronyms, add full name). Also, homogenize the acronyms of UTI throughout the text (if the description of urinary tract infections is already included with their respective acronyms, simply add UTI in the following paragraphs of the text).
Indicate the bibliographic references to the interventions developed.
Add the full name: Consolidated criteria for reporting qualitative research (COREQ).
Results
The wording should be improved.
The sociodemographic characteristics of the participants should be described.
Finally, establish a table of the main results found in the study.
Discussion
Improve the comparison of the results with published studies. In addition, make recommendations for future studies.
Author Response
Please find the point by point reply in the attachment.

Reviewer 2 Report
Comments and Suggestions for Authors
Schuster et al presented an interesting manuscript regarding barriers to implementing strategies to improve the treatment of urinary tract infections among GPs.
They report the results of 19 semi-structured online interviews describing the GPs’ feedback on a multimodal improvement intervention.
Overall, GPs feel that interventions were positive and could help to improve their prescription practice. It was also possible to identify some barriers to change, mostly related to the impression that GPs’ practice did not significantly influence the problem of bacterial resistance and lack of time or available resources.
I think this is an interesting paper reporting on a very important issue: behavioral changes in antibiotic prescription.
However, I also have some concerns mostly related to the presentation format.
1- The manuscript can be largely resumed. For instance, the introduction described the importance of urinary tract infection, which is not in the scope of this work.
2- The results section should be largely summarized. The manuscript is not intended to be a report of the intervention. One or two tables could be used to summarize the results. Direct quotations of the interviews could be reported in the tables or even in Supplementary files. Using those in the text leads to an unnecessarily long manuscript.
3- Semi-quantitative information would also help (for instance, 9/19 reported to appreciate; 5/19 did not). Although I understand that these are very small numbers, this could help to understand the rate of acceptance of the different measures.
4- Discussion could also be summarized. It largely repeats the results and comments on the published observations. I would like to see some comments on comparisons between authors and published findings.
Author Response

(The authors gave the same response as above.)

Round 2
Reviewer 1 Report
Comments and Suggestions for Authors
The observations I made were analyzed and appropriately addressed by the authors of the article. This process of paying attention to the observations has substantially contributed to raising the scientific quality of the paper. The improvements implemented reflect a serious commitment to methodological rigor and scientific accuracy, thus strengthening the credibility and value of the study.
Reviewer 2 Report
Comments and Suggestions for Authors
I think the manuscript is very much improved. I have no further comments.